# Fixed-Bed Studies of Landfill Leachate Treatment Using Chitosan-Coated Carbon Composite

**Fatima Batool [1,†], Tonni Agustiono Kurniawan [1,2,\*,†], Ayesha Mohyuddin [1], Mohd Hafiz Dzarfan Othman [3], Abdelkader Anouzla [4], Christia Meidiana [5], Hui Hwang Goh [6] and Kit Wayne Chew [7]**

[1] Department of Chemistry, School of Science, University of Management and Technology, Lahore 54770, Pakistan
[2] College of Environment and Ecology, Xiamen University, Xiamen 361102, China
[3] Advanced Membrane Technology Research Centre (AMTEC), Faculty of Chemical and Energy Engineering, Universiti Teknologi Malaysia (UTM), Skudai 81310, Johor Bahru, Malaysia
[4] Department of Process Engineering and Environment, Faculty of Science and Technology, University Hassan II of Casablanca, Mohammedia 28830, Morocco
[5] Department of Regional and Urban Planning, Faculty of Engineering, Brawijaya University, Malang 65145, Indonesia
[6] School of Electrical Engineering, Guangxi University, Nanning 530004, China
[7] School of Chemistry, Chemical Engineering and Biotechnology, Nanyang Technological University (NTU), Singapore 637459, Singapore
\* Correspondence: tonni@xmu.edu.cn
† The 1st and the 2nd authors equally contribute to this work and mutually share the first authorship.

**Abstract:** The feasibility of a chitosan-coated coconut-shell (CS) carbon composite for landfill leachate treatment in a fixed-bed study was investigated in terms of COD and $NH_3$-N removal. The surface of the composite was characterized using SEM, FT-IR, and XRD to assess any changes before and after column operations. To enhance its cost-effectiveness, the saturated composite was regenerated using NaOH. The results showed that the composite had significantly better removal of both COD and $NH_3$-N, as compared to CS and/or chitosan ($p \leq 0.05$; ANOVA test), respectively. The breakthrough curve obtained from the fixed-bed studies exhibited an ideal "S" shape. The breakthrough points for the adsorbents followed the order of CS at BV 76 < chitosan at 200 BV < composite at BV 305. It was also found that a low flow rate and deeper bed depth of the packed adsorbent were necessary for achieving optimal column operations. The composite achieved 96% regeneration in the first cycle. However, even with the enhanced adsorption of target pollutants by the composite through chitosan coating, the treated effluents still could not meet the required COD and $NH_3$-N effluent limits of less than 200 and 5 mg/L, respectively, as mandated by legislation. Nonetheless, the findings suggest that low-cost composites derived from unused resources can be employed as effective adsorbents for wastewater treatment.

**Keywords:** adsorption; greenhouse gas; landfill leachate; low-cost adsorbent; resource recovery

## 1. Introduction

In recent years, urbanization in emerging economies has led to increased population growth in cities, resulting in urban growth and interconnectivity [1]. A significant proportion of migrant workers, about 50%, reside in cities, and their population has been increasing by 2% annually [2]. Recent projections indicate that the global population has exceeded eight billion people in November 2022, with India surpassing China to become the world's most populous country [3].

By 2050, it is projected that 70% of the global population will live in cities, resulting in the generation of about 1.3 billion metric tons (Mt) of municipal solid waste (MSW) annually [4]. Consequently, global waste generation is expected to increase significantly. Various industries and sectors worldwide generate nearly 10 billion metric tons (Mt) of waste each year [5].

The continuous growth in waste disposal in landfills contributes to over 2% of $CO_2$ emissions from the urban sector [6]. In addition to leachate, landfills produce a substantial amount of landfill gas (LFG) [7]. Bacterial decomposition of the organic-waste fraction in landfills produces $CH_4$ at a rate of about 75%, along with $CO_2$ (20%), water, and other trace constituents. $CH_4$ is a potent greenhouse gas with a global warming potential that is 25 times higher than that of $CO_2$ [8].

The current approach to solid-waste management (SWM) in developing countries, particularly in relation to municipal solid waste (MSW), is characterized by a linear and noncyclical model. Landfilling is the predominant method of waste disposal globally [9]. Within landfills, the waste undergoes anaerobic degradation through physico-chemical and biological processes, resulting in the generation of landfill leachate.

In this linear paradigm, waste is not viewed as a valuable resource that can be integrated into sustainable business models, such as composting or energy generation. Instead, it is perceived as a burden that needs to be eliminated [10]. Consequently, waste is rarely considered as part of a production–consumption–recovery cycle or evaluated from an environmental sustainability perspective. This linear approach poses challenges, as the planet lacks the carrying capacity to effectively manage the daily waste generated by its inhabitants, particularly with regard to plastic waste, which contributes to irreversible environmental damage [11]. It is projected that the number of people affected by organic pollution will increase by 108%, from 1.2 billion in 2000 to 2.5 billion in 2050 [12]. Wastewater discharged from urban areas can be a major source of pollution in rivers and underlying groundwater [13].

The global water-shortage crisis is exacerbated by economic development, population growth, and increasing urbanization. According to the United Nations (UN), about one-third of the world's population lacks access to clean and safe drinking water, and approximately 60% lack access to sanitation facilities [14]. Sanitation-related diarrheal diseases result in the loss of thousands of children's lives every day [15].

Cities are significant contributors to $CO_2$ emissions, accounting for around 70% of global emissions [16]. However, cities also face challenges in providing waste infrastructure services in a carbon-neutral and sustainable manner, often due to limited public budgets [17]. To achieve the goals outlined in the 2030 UN Sustainable Development Goals (SDGs), urban development needs to prioritize environmental sustainability and utilize advanced water technologies to address issues such as landfill leachate and wastewater generated from the decomposition of organic waste in landfills [18].

Leachate is formed when water percolates through waste layers in landfills, carrying organic and inorganic components from the waste [19]. Leachate from young landfills contains high concentrations of organic materials that pollute the surrounding environment, including soil and groundwater [20]. The quantity of leachate produced is directly influenced by precipitation levels around the landfill. Depending on the landfill's characteristics and the waste it contains, leachate can be hazardous and toxic [21]. Typically, leachate has a high biochemical oxygen demand (BOD) and high concentrations of organic carbon, indicated by chemical oxygen demand (COD) and $NH_3$-N, which are considered refractory pollutants due to their toxicity to organisms [22]. In young leachate, COD concentrations range from 20,000 to 60,000 mg/L, whereas in stabilized landfills, the COD strength ranges from 5000 to 15,000 mg/L. The $NH_3$-N concentration in young leachate ranges from 5000 to 10,000 mg/L, whereas in older landfills, it ranges from 500 to 4500 mg/L [23,24].

Landfill leachate poses significant environmental risks and has a negative impact on water quality, limiting the use of resources for various purposes and harming ecosystems [25]. The cost of recovering from the impacts of leachate can be substantial, with an annual loss of over USD 4 billion in the U.S. due to leachate infiltration into groundwater [26]. Modern sanitary landfills are designed with liners to prevent leachate from infiltrating groundwater, and the collected leachate is treated to mitigate its pollutants [27]. Safe and environmentally friendly treatment of landfill leachate has become an urgent issue [28].

The perception of landfill leachate has shifted from it being viewed as waste to being a valuable source of resources, including nutrients, energy, and water [29]. Recovering these resources is of increasing interest to reduce greenhouse-gas emissions and optimize resource utilization [30]. However, careful analysis is required to ensure that resource recovery does not result in increased energy use and greenhouse-gas emissions compared to conventional water treatment methods [30].

Landfill leachate treatment also contributes to climate change, as it generates greenhouse gas (GHG) emissions, primarily $CH_4$, due to the decomposition of organic matter [31,32]. Globally, a significant portion of wastewater is discharged into the environment without adequate tertiary treatment, posing health and environmental risks [33]. Novel treatment technologies are being developed to reduce the environmental risks associated with refractory pollutants in landfill leachate [34].

Various methods for landfill leachate treatment have been tested, including physicochemical treatments [35], advanced oxidation processes [36], and biological processes [37]. Traditional leachate treatments consume less energy than the energy that can theoretically be extracted from the carbon within the wastewater [38,39]. Biological solutions may be economically attractive, but their stability in storm-prone areas and extreme weather conditions caused by climate change is a concern [40]. Chemical precipitation is reliable for leachate treatment, although it generates sludge that requires extra treatment costs [41].

A preliminary study using coconut-shell (CS)-based activated carbon as an adsorbent for COD and/or $NH_3$-N removal from solutions in batch mode has been conducted [42]. However, continuous-adsorption systems, such as fixed-bed columns, are more efficient for treating larger effluent volumes in a shorter time. Fixed-bed column studies provide valuable insights into the dynamic process, scale-up potential, and mass-transfer phenomena related to pollutant–material interactions [43]. Therefore, there is a growing need to conduct fixed-bed studies before scaling up landfill leachate treatment processes [44].

To demonstrate its novelty, this study focuses on exploring the potential of a chitosan-coated carbon composite for the treatment of landfill leachate, specifically targeting the removal of COD and/or $NH_3$-N. The changes in the composite's characteristics before and after the column studies were investigated through SEM, FT-IR, and XRD analyses. To ensure the cost-effectiveness of the treatment process, the spent adsorbent was regenerated using NaOH. The aim of this work is to provide users with an efficient and affordable solution for leachate treatment, enabling them to meet the discharge limit mandated by legislation.

## 2. Material and Methods

### 2.1. Materials

The properties of the coconut shell (CS) and chitosan used in this study are presented in Table 1. Raw leachate was collected from a local open dump (Figure 1).

**Table 1.** Properties of CS.

| Property | |
|---|---|
| Packing density (g/cm$^3$) | 0.65 |
| Total surface area (m$^2$/g) | 15 |
| Solid density (g/cm$^3$) | 0.57 |
| Particle size (mm) | 0.35 |
| Pore volume (mL/g) | 0.04 |

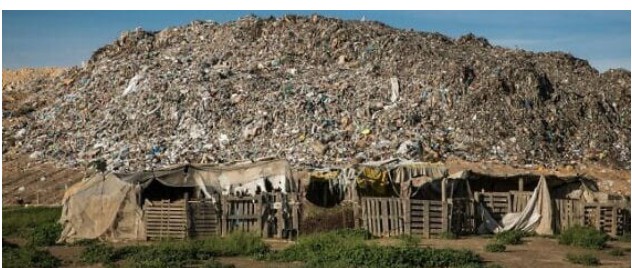

**Figure 1.** Local open dump.

With a maximum capacity of $24 \times 10^6$ m$^3$ and a total land of 25 ha, the landfill receives about 2500 Mt of MSW per day from the city's population of 11 million in 2023 [45]. Since its first operation in 2015 [46], the dominant waste dumped in the landfill has been refuse originating from household, construction, and industrial activities. The landfill generates about 1500 m$^3$ of leachate daily (Figure 2) [47].

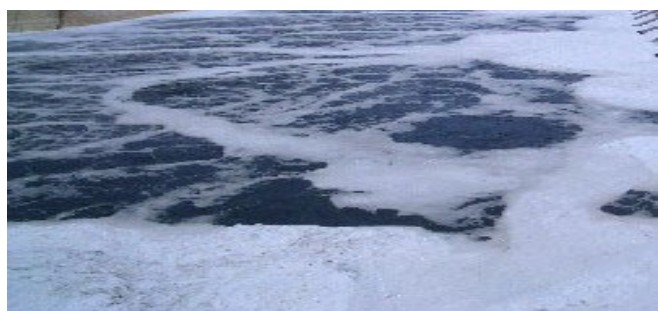

**Figure 2.** Stabilized leachate.

The leachate samples were collected and transferred to 20 L polyethylene carboys, and then stored in a dedicated chamber at a temperature of 4 °C to prevent any alterations in their physicochemical properties prior to the column studies. The collected samples were promptly analyzed using the Standard Methods [48] for several parameters, including BOD$_5$, pH, COD, TOC, NH$_3$-N, alkalinity (as CaCO$_3$), and conductivity.

The pH of the leachate solution was determined using an Orion 7A model pH meter (Pakistan), and pH adjustment was carried out using 0.1 M HCl and/or 0.1 M NaOH solutions. The concentrations of COD and NH$_3$-N were measured using a spectrophotometer of UH415AD type (Tokyo, Japan). The conductivity of the samples was determined using a conductivity meter of Lutron C303 type (Shanghai, China), whereas the TOC content was analyzed using a TOC analyzer of Shimadzu 500 type (Torrance, CA, USA)

### 2.2. Characterization of the Composite

The morphology of the chitosan-coated CS composite was examined before and after treatment using scanning electron microscopy (SEM) coupled with energy-dispersive X-ray spectroscopy (EDX). The SEM-EDX analysis was conducted using a Hitachi SU830 Regulu model (Tokyo, Japan). This allowed for the visualization and elemental analysis of the composite's surface, providing information about its microstructure and composition [49].

The crystallinity of the composite was analyzed using a Shimadzu 300 model diffractometer (Tokyo, Japan). For this analysis, the composite samples were powdered and subjected to X-ray diffraction (XRD) measurements. The XRD analysis was performed in the 5 to 45° range with a step size of 0.02° (2θ). Cu was used as the X-ray source, with a voltage of 30 kV and a current of 30 mA. This analysis provided insights into the crystallinity of the structure and phase composition of the composite material [50].

Fourier transform infrared (FT-IR) characterization was carried out using a spectrophotometer in the wavelength range from 4000 to 400 cm$^{-1}$, with a resolution of 4 cm$^{-1}$. This

analysis allowed for the identification of functional groups present in the composite material based on the characteristic absorption bands in the infrared spectrum. It provided information about the chemical bonds and molecular structure of the composite.

### 2.3. Fixed-Bed Studies

The column studies were conducted using a glass column with an internal diameter of 1 cm and a height of 30 cm, as shown in Figure 3. The column was filled with 10 g of the chitosan-coated carbon composite as the adsorbent, creating a bed height of 5 cm. To secure the adsorbent bed, glass beads with a diameter of 1 cm were placed at the top and bottom layers of the column.

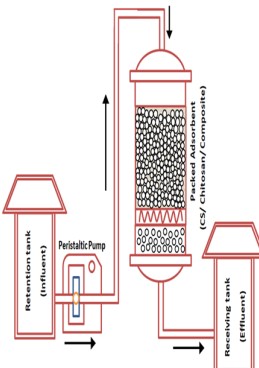

**Figure 3.** Fixed-bed study of leachate treatment.

The leachate samples, totaling 2 L, were introduced into the column in an upward flow mode. The flow rate varied during the experiments, ranging from 3 L/min. The treated effluents were collected from the outlet positioned at the top of the column. The pH of the solution was continuously monitored and maintained stable throughout the experiments. During the column operations, aliquots of the treated effluent were collected at different time intervals for further analysis. The purpose of the intervals was to evaluate the effectiveness of the chitosan-coated carbon composite in removing contaminants from the leachate and to assess the performance of the column system.

The breakthrough curves were determined by plotting $C_e/C_o$ against time. The concentration of the remaining pollutants was calculated based on the absorbance in a UV-Vis spectrophotometer at 600 nm (for COD) and 626 nm (for $NH_3$-N) [51], which are the maximum absorbances of both target pollutants.

### 2.4. Regeneration Tests

To improve its cost-effectiveness during operation, spent adsorbent was regenerated with NaOH. Regeneration (%) of the composite was measured based on Equation (1) [51]:

$$RE(\%) = (R_t/R_0) \times 100\% \tag{1}$$

where: $R_t$ and $R_0$ are final and initial adsorption capacities of the composite, respectively.

### 2.5. Statistical Analysis

To ensure the reliability and accuracy of the data, the column studies were conducted in triplicate, meaning that the experiments were repeated three times under identical conditions. This helped to minimize experimental variability and provides a more robust dataset. The mean values of the data sets were then calculated and presented, which allowed for a more representative understanding of the overall performance of the chitosan-coated carbon composite in removing contaminants from the leachate [52].

The accuracy of the data was considered ideal, as indicated by the standard deviation of their removal being less than 1.0% [53]. A low standard deviation suggests that the

results were consistent and reliable across the replicated experiments and indicates a high level of precision in the measurements. This strengthens the credibility of the data and reinforces the validity of the conclusions drawn from the study [54].

## 3. Results and Discussion

### 3.1. Properties of Leachate Samples

The properties of the leachate samples from the local landfill are presented in Table 2.

**Table 2.** Characteristics of leachate samples.

| Parameter | Value |
|---|---|
| pH | 7.8 |
| $BOD_5$ (mg/L) | 500 |
| COD (mg/L) | 7000 |
| $BOD_5$/COD | 0.07 |
| TOC (mg/L) | 2100 |
| $Cr^{3+}$ (mg/L) | 21.l2 |
| $Na^+$ (mg/L) | 11.01 |
| $Ca^{2+}$ (mg/L) | 12.51 |
| $NH_3$-N (mg/L) | 1500 |
| Conductivity (mS/cm) | 11.5 |
| Alkalinity (as $CaCO_3$) (mg/L) | 10,500 |

The results presented in Table 2 indicate that the leachate belonged to a stabilized landfill, as evidenced by the low biodegradability ratio ($BOD_5$/COD) of less than 0.1. This suggests that a significant portion of the organic fraction in the waste had already undergone degradation, mainly producing $CH_4$. In such cases, physicochemical treatments are more suitable than biological processes, despite the latter being more cost-effective.

Another observation from the analysis is the presence of conductivity in the samples. This indicates the presence of inorganic pollutants, such as heavy metals, in the leachate. The presence of these pollutants suggests that there might be competition between the refractory pollutants and heavy metals for adsorption sites on the surface of the chitosan-coated carbon composite. This competition can potentially affect the overall adsorption capacity and efficiency of the composite in removing contaminants from the leachate [55].

### 3.2. Characterization of the Composite

The morphology of the chitosan-coated carbon composite was analyzed before and after the column studies, and the results are presented in Figure S1. The images provide visual information about the structural characteristics and changes in the composite [56].

The FTIR spectra (Figure S2) show several characteristic absorption bands that reveal the molecular composition of the composite. The absorption band at 3410 cm$^{-1}$ corresponds to the stretching vibration of -OH groups, indicating the presence of lignocellulosic materials in the composite due to the CS [57]. The band at 2920 cm$^{-1}$ indicates the existence of $CH_3$ groups in the adsorbent. A sharp carbonyl stretch appeared at 1730 cm$^{-1}$, suggesting the presence of ketone and carboxylic acid groups in the molecular structure of the composite. This indicates the potential for functional groups to interact with pollutants in the leachate [57].

Another spectrum shows a broad band at 1600 cm$^{-1}$, which suggests the presence of C=C bonds from aromatic rings. Additionally, there are stretches observed at 1245 and 1050 cm$^{-1}$, which are derived from the C-O vibrations of C$sp^2$ and C$sp^3$, respectively. These vibrations provide further information about the composition and structure of the composite. Overall, the FTIR analysis helps to characterize the molecular components of the chitosan-coated composite and provides insights into its interactions with the leachate contaminants [58].

### 3.3. Fixed-Bed Studies

Column studies are crucial for evaluating the performance of adsorbents in practical applications. Fixed-bed studies are commonly used in industry because of their continuous operation and ease of use. These studies provide valuable information on important parameters, such as breakthrough time, flow rate, and loss of adsorption capacity throughout the cycles [59]. The operation of the column is essential for industrial-scale applications as it provides reliable data on the behavior of the adsorbent. By analyzing the breakthrough curve, users can assess the adsorbent's capacity to bind a specific adsorbate and evaluate its practical usability and viability for industrial applications [60].

During column studies, two important aspects of the treatment process are the recovery of adsorbate material and the renewal of the adsorbent. It is necessary to determine the adsorption capacity and kinetics of the adsorbents to design effective column operations. One approach to obtaining this information is by examining the ratio of the number of bed volumes (BV) that can be treated by the adsorbent until it becomes completely exhausted due to the amount of adsorbate in the wastewater [61].

The breakthrough graph provides insights into the impurity adsorption behavior on the surface of the adsorbent and helps in understanding the adsorption process. The findings from fixed-bed studies of various adsorbents are presented in Table 3, which provides a comprehensive overview of their performance in terms of adsorption capacity and breakthrough behavior [62].

**Table 3.** $NH_3$-N and COD removal by all adsorbents (CS, chitosan, and composite).

| Adsorbent | Pollutant | Concentration (mg/L) | Breakthrough Point (h) | BV at Breakthrough Point | Saturated Point (h) | BV at Saturated Point | Volume of Treated Influent (L) |
|---|---|---|---|---|---|---|---|
| CS | | 7000 | 10 | 76.4 | 50 | 458.5 | 10.8 |
| Chitosan | COD | 7000 | 25 | 200 | 80 | 611 | 14.4 |
| Composite | | 7000 | 40 | 305.7 | 120 | 917 | 21.6 |
| CS | | 1500 | 9 | 68.7 | 49 | 374 | 8.8 |
| Chitosan | $NH_3$-N | 1500 | 24 | 183.4 | 79 | 603 | 14.2 |
| Composite | | 1500 | 39 | 298 | 119 | 909 | 21.4 |

Note: BV: bed volume

### 3.3.1. Breakthrough

The breakthrough point is the point at which a specific amount of the influent is detected in the effluent, indicating that the adsorbent's capacity to remove the impurity has reached its limit. The number of bed volumes (BV) is the ratio between the volume of the adsorbate solution and the amount of adsorbent used [63]. The two parameters are commonly used to compare the ability of different adsorbents to remove a specific impurity from the influent [64].

In the fixed-bed study, the operating capacity was determined based on the breakthrough point, which is typically defined as the point at which 5% of the influent concentration ($C_o$) is detected in the effluent ($C_e$). Column experiments were conducted for all adsorbents, including coconut shell (CS), chitosan, and the composite material, to evaluate their practical effectiveness in removing COD and $NH_3$-N from the leachate. Figure 4 illustrates a typical breakthrough curve, which plots the ratio of influent concentration ($C_o$) to effluent concentration ($C_e$) against the number of bed volumes (BV) passing through the column until it completely exhausts the adsorbent's capacity. The breakthrough curve provides important information about the adsorption behavior and the efficiency of the adsorbent, allowing for the assessment of its performance in removing the targeted impurities from the leachate [65].

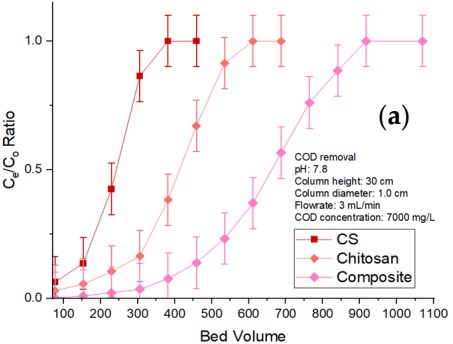
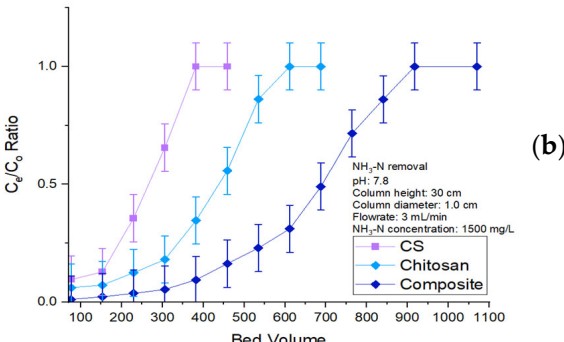

**Figure 4.** Breakthrough curves of all adsorbents on (**a**) COD and (**b**) NH$_3$-N removal for the first run.

In Figure 4, the breakthrough curves for all adsorbents exhibit a typical "S" shape. It is interesting to note that although the equilibrium point is reached when there is no concentration difference between the influent and effluent (Ce/Co = 1), the breakthrough threshold is achieved when COD and NH$_3$-N first appear in the effluent (C$_e$/C$_o$ = 0.05).

For the coconut shell (CS), the breakthrough points for both COD and NH$_3$-N removal occur at 76.4 bed volumes (BV) or 1.8 L of influent. The breakthrough quickly progresses at the start of adsorption but declines steadily until reaching complete exhaustion at 458.5 BV (10.8 L). This indicates that CS quickly removes COD and NH$_3$-N in the initial stages of adsorption when there are ample available adsorption sites on the adsorbent's surface. However, as the surface sites become saturated, the removal rate slows down. The breakthrough curve of CS follows the ideal "S"-shape profile, indicating efficient use of the adsorbent and a large adsorption zone within the CS bed [66].

In comparison, chitosan achieves the breakthrough point later at 200 BV (4.9 L) and becomes completely exhausted at 687 BV (approximately 16.2 L) for both COD and NH$_3$-N removal. The composite material reaches the breakthrough point even later at 305.7 BV (7.2 L) and becomes fully saturated at 917 BV (21.6 L). The significant differences in COD and NH$_3$-N removal among the adsorbents suggest that the composite exhibits the best performance compared to CS and chitosan. These results indicate that the composite material has superior removal capabilities for COD and NH$_3$-N compared to CS and chitosan, as evidenced by the delayed breakthrough point and higher saturation capacity [67].

The maximum column bed capacity ($q_{total}$) for the COD and NH$_3$-N concentrations and flow rates was determined based on Equation (2) [68]:

$$q_{total} = \frac{QA_c}{1000} = \frac{Q}{1000} \int_{t=0}^{t_{total}} C_{ads} dt \qquad (2)$$

where: $Q$ is the flow rate (L/h), and $QA_c$ is the area under the breakthrough curve, whereas $q_{total}$ and $C_{ads}$ represent the maximum bed ability (mg) and adsorbed COD and NH$_3$-N concentrations (mg/L), respectively. The value of $q_{e(exp)}$ was calculated based on [68]:

$$q_{e(exp)} = \frac{q_{total}}{m} \qquad (3)$$

where: $m$ (g) is the amount of the adsorbent (CS, chitosan, and composite) packed in the column.

Table 4 presents the calculated values of the total adsorption capacity ($q_{total}$) and the experimental adsorption capacity ($q_{exp}$) for coconut shell (CS), chitosan, and the composite material. The adsorption capacities were determined by calculating the area ($A$) under the breakthrough curve until reaching complete exhaustion. The experimental $q_{exp}$ values were obtained from the dynamic study by dividing the maximum bed ability by the total mass of the respective adsorbent. The values provide an estimation of the adsorption capacity based on the experimental data [69].

**Table 4.** Parameters calculated from breakthrough curves.

| Adsorbents | Q (mL/min) | m (g) | COD | | | NH₃-N | | |
|---|---|---|---|---|---|---|---|---|
| | | | A | $q_{total}$ (mg) | $q_{exp}$ (mg/g) | A | $q_{total}$ (mg) | $q_{exp}$ (mg/g) |
| CS | 3 | 10 | 1807.1 | 5.421 | 0.542 | 1958.9 | 5.8767 | 0.587 |
| Chitosan | 3 | 10 | 3107.02 | 9.321 | 0.932 | 3178.01 | 9.5340 | 0.953 |
| Composite | 3 | 10 | 5041.09 | 15.12 | 1.512 | 5106.1 | 15.318 | 1.531 |

### 3.3.2. Effects of Flow Rate

In Figure 5, the breakthrough curves of the composite are presented for different flow rates (3, 9, and 15 mL/min) and influent concentrations of NH₃-N (1500 mg/L) or COD (7000 mg/L). The goal of this investigation was to examine the behavior of the composite under different flow-rate conditions and high concentrations of NH₃-N and COD.

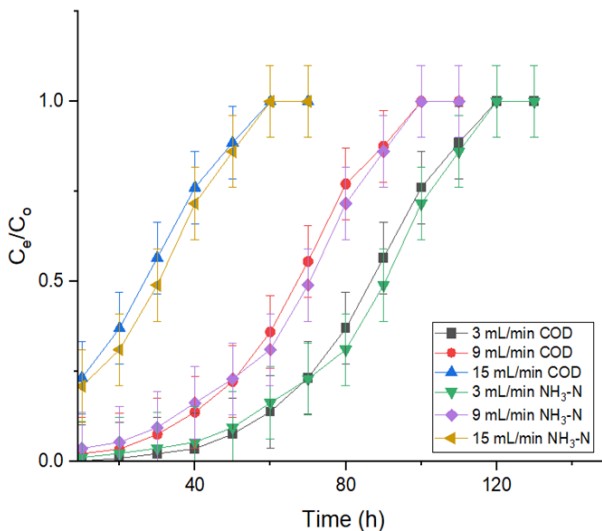

**Figure 5.** Effect of flow rate on the composite's adsorption of COD and NH₃-N.

By varying the flow rate, the physicochemical reactions taking place in the dynamic adsorption process can be influenced. This, in turn, affects the adsorption equilibrium between the composite material and NH₃-N or COD. The vigorous behavior of the composite's breakthrough curves provides insight into its performance under different flowrate conditions and high pollutant concentrations [70].

Based on the information provided, Figure 5 shows the breakthrough curves of the composite material at different flow rates (3, 9, and 15 mL/min) for NH₃-N and COD removal. The time required to reach the breakthrough point ($C_e/C_0 = 0.05$) varied depending on the flow rate. At a flow rate of 3 mL/min, it took approximately 50 h to reach the breakthrough point, whereas at 9 mL/min, it took about 30 h. The highest flow rate of 15 mL/min resulted in a faster breakthrough, with the curve reaching the breakthrough point in less than 24 h.

The results indicate that higher flow rates led to faster breakthroughs of NH₃-N and COD, suggesting that there were fewer physicochemical interactions between the composite and the adsorbates in the reactor at higher fluid flow rates. On the other hand, a lower flow rate was preferred during the column operations to maximize the treated bed volume (BV) because a longer contact time was needed for NH₃-N and COD adsorption. A longer contact time allows for more physicochemical interactions between the composite and the adsorbates, increasing the removal efficiency. Additionally, a lower flow rate enhances the mass-transfer resistance, providing a stronger driving force for mass transfer in the fixed-bed reactor.

The findings suggest that flow rate, along with the properties of the influent, is one of the major factors affecting the removal of $NH_3$-N and COD by the composite material in the reactor. Controlling the flow rate can help to optimize the adsorption process and maximize the efficiency of the $NH_3$-N and COD removal.

### 3.3.3. Effects of Bed Depth

Based on the given information, Figure 6 shows the effect of bed depth on $NH_3$-N and COD removal when using the composite material in a closed system. The experiments were conducted with a constant flow rate of 3 mL/min and influent $NH_3$-N and COD concentrations of 1500 and 7000 mg/L, respectively.

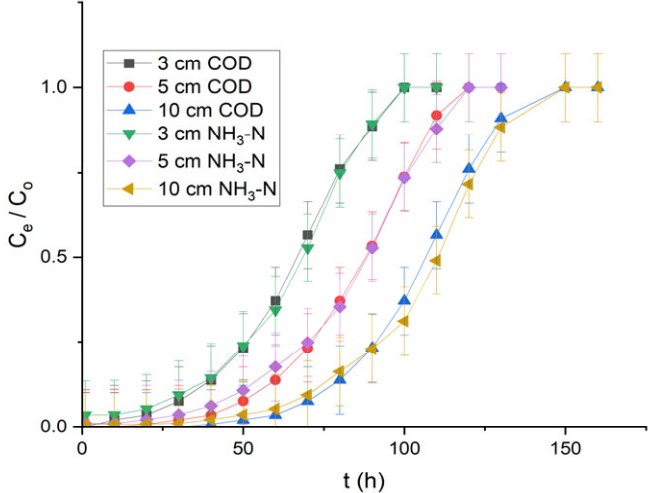

**Figure 6.** Effect of bed depth on composite's adsorption of COD and $NH_3$-N.

The results indicate that as the bed depth increased from 3 to 10 cm, the time required for the composite material to reach the breakthrough point increased by 40%, from 20 to 60 h. This suggests that increasing the bed depth provides more time for the composite material to interact with the $NH_3$-N in the solution, resulting in a higher removal rate.

At deeper bed depths, $NH_3$-N molecules have a longer contact time with the composite material, allowing for increased physicochemical interactions and enhanced adsorption. The deeper bed depths also provide a larger surface area for $NH_3$-N adsorption, offering more binding sites for the adsorbate. As a result, the removal efficiency of $NH_3$-N is improved. It is important to note that the bed depth is a critical parameter in fixed-bed adsorption systems, as it directly influences the contact time between the adsorbent and the adsorbate. By adjusting the bed depth, the adsorption capacity and efficiency of the composite material for $NH_3$-N and COD removal can be optimized.

### 3.4. Regeneration

In the context of adsorption processes, regeneration refers to the steps taken to restore the adsorbent's capacity after it has become saturated with $NH_3$-N and COD. Regeneration is important for sustaining the cost-effectiveness of wastewater treatment applications using fixed-bed reactors. Regeneration typically involves desorption, which is the initial phase of regeneration. Desorption refers to the removal of the adsorbate ($NH_3$-N and COD) from the surface of the adsorbent. This step aims to release the adsorbed pollutants from the adsorbent, making it available for further use in subsequent column runs.

Desorption can be achieved through various methods, such as changing the pH or temperature of the system, using a desorbing agent or solvent, or applying a combination of techniques. The choice of desorption method depends on the nature of the adsorbate, the adsorbent material, and the specific requirements of the wastewater treatment process.

Once desorption is complete, the regenerated adsorbent can be reused in the fixed-bed reactor for further adsorption cycles. By effectively regenerating the spent adsorbent, the

overall operational cost of the wastewater treatment process can be reduced, making it economically viable for long-term applications.

The desorption and regeneration of $NH_3$-N and COD by the composite, CS, and chitosan were investigated and compared. Figure 7 shows that the composite required a higher volume of regenerant and a longer desorption cycle to complete the desorption of $NH_3$-N and COD compared to CS. This indicates that the composite had a stronger adsorption capacity and a higher affinity for $NH_3$-N and COD, resulting in a more efficient removal process. However, the higher regenerant volume required for the composite indicates a higher operational cost in terms of regenerant consumption.

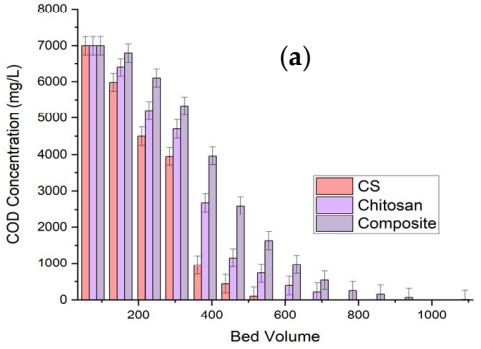 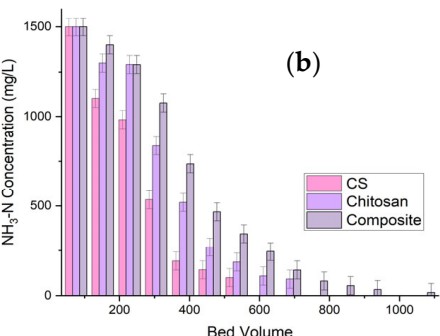

**Figure 7.** Regeneration curves of CS, chitosan, and composite for (**a**) COD and (**b**) $NH_3$-N.

Table 5 summarizes the regeneration effectiveness of the adsorbents. It shows that after the first regeneration cycle, the composite achieved recovery of approximately 95–96% of the $NH_3$-N and COD. This indicates that the composite retained its adsorption capacity and effectiveness even after undergoing regeneration. The regeneration process was able to restore a significant portion of the adsorbent's original adsorption performance, making it suitable for multiple adsorption–desorption cycles. Overall, the findings suggest that the composite exhibited superior adsorption and regeneration characteristics compared to CS and chitosan, providing a promising solution for efficient $NH_3$-N and COD removal in wastewater treatment applications.

**Table 5.** Regeneration of CS, chitosan, and composite for the first cycle.

| | COD | | | | $NH_3$-N | | | |
|---|---|---|---|---|---|---|---|---|
| Types of Adsorbents | Before First Regeneration (mg/g) | After First Regeneration (mg/g) | *RE* (%) | *LAC* (%) | Before First Regeneration (mg/g) | After First Regeneration (mg/g) | *RE* (%) | *LAC* (%) |
| CS | 0.542 | 0.480 | 88 | 12 | 0.587 | 0.501 | 85 | 15 |
| Chitosan | 0.932 | 0.850 | 91 | 9 | 0.953 | 0.863 | 90 | 10 |
| Composite | 1.512 | 1.460 | 96 | 4 | 1.531 | 1.465 | 95 | 5 |

Note: RE: Regeneration efficiency; LAC: loss of adsorption capacity.

### 3.5. Thomas Model

The Thomas model surmises that there are no external or internal diffusion limitations affecting the adsorption process in the column. It provides a mathematical description of the adsorption kinetics and can be used to assess the efficiency of the column in terms of the composite's adsorption capacity. By fitting the experimental data to the Thomas model, the maximum adsorption capacity ($q_e$) and the rate constant of adsorption ($k_{th}$) can be determined. These parameters provide insights into the adsorption performance and kinetics of the composite in the column operation. It is important to note that the Thomas model's assumptions may not hold true in all cases, especially in complex systems. However, it serves as a useful tool for analyzing and evaluating the column efficiency based on the composite's adsorption capacity.

The Thomas model's kinetics are shown in Equation (4):

$$ln\left(\frac{C_o}{C_t} - 1\right) = \frac{K_{th}q_{cal}m}{Q} - K_{th}C_o t \tag{4}$$

where: $k_{th}$ represents the Thomas rate constant (L/h·mg), $q_{cal}$ is the maximum equilibrium of the adsorbate (mg/g), and $Q$ is the flow rate (L/h).

Figure 8 and Table 6 demonstrate the application of the Thomas model to the experimental data for CS, chitosan, and composite adsorbents at a flow rate of 3 mL/min. The linear plots in Figure 8 represent the adsorption kinetics based on the Thomas model.

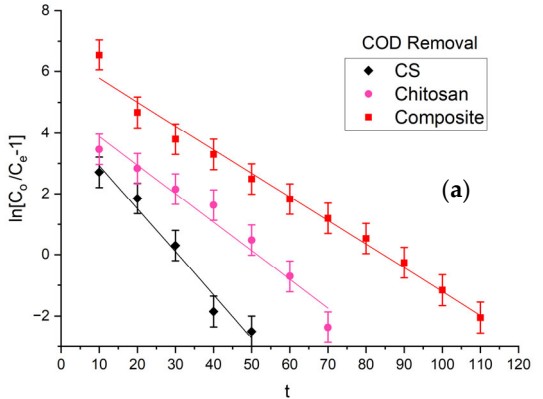 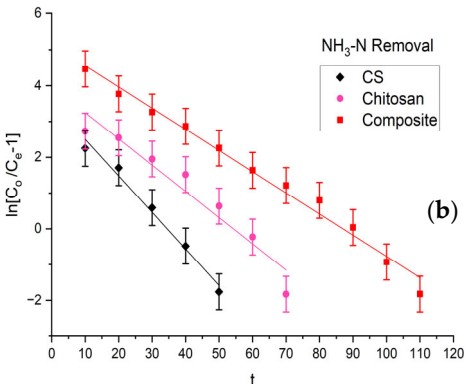

**Figure 8.** Kinetic plots of Thomas model for all adsorbents for (**a**) COD and (**b**) NH$_3$-N adsorption. (COD: 7000 mg/L; NH$_3$-N: 1500 mg/L, 25 °C, 3 mL/min of flow rate).

**Table 6.** Thomas model's kinetics of CS, chitosan, and composite for COD and NH$_3$-N adsorption.

| Pollutant | Type of Adsorbent | $C_o$ (mg/L) | Flow Rate (mL/min) | Bed Depth (cm) | $q_{o,cal}$ (mg/g) | $q_{o,exp}$ (mg/g) | $K_{th}$ ($\times 10^3$) (L/h·mg) | $R^2$ |
|---|---|---|---|---|---|---|---|---|
| | Composite | 1500 | 3 | 5 | 1.338 | 1.531 | −0.4035 | 0.9845 |
| NH$_3$-N | Chitosan | 1500 | 3 | 5 | 0.854 | 0.953 | −0.5571 | 0.9784 |
| | CS | 1500 | 3 | 5 | 0.488 | 0.587 | −0.8419 | 0.9180 |
| COD | Composite | 7000 | 3 | 5 | 1.317 | 1.512 | −0.1097 | 0.982 |
| | Chitosan | 7000 | 3 | 5 | 0.875 | 0.932 | −0.1412 | 0.975 |
| | CS | 7000 | 3 | 5 | 0.432 | 0.542 | −0.2302 | 0.922 |

The $R^2$ values provided in Table 6 indicate the goodness of fit between the experimental data and the Thomas model. The $R^2$ values ranging from 0.918 to 0.9845 suggest that the experimental data closely follow the predicted behavior of the Thomas model. A higher $R^2$ value indicates a better fit between the model and the experimental data.

Additionally, Table 6 provides the values of $K_{th}$ (rate constant of adsorption) and $q_{cal}$ (calculated adsorption capacity) for each adsorbent. These values are obtained through the fitting of the experimental data to the Thomas model. The rate constant of adsorption ($K_{th}$) represents the speed at which adsorption occurs, whereas the calculated adsorption capacity ($q_{cal}$) represents the maximum amount of adsorbate that can be adsorbed onto the adsorbent's surface. By analyzing the $R^2$ values, $K_{th}$, and $q_{cal}$, one can assess the applicability of the Thomas model to determine the adsorption kinetics of CS, chitosan, and composite adsorbents in the column operations.

According to the information provided, Table 5 indicates that the bed capacity ($q_{cal}$) of the composite decreased for the adsorption of both NH$_3$-N and COD, as evidenced by the values of $K_{th}$. The decrease in $q_{cal}$ suggests that the adsorption capacity of the composite decreased over successive desorption and regeneration cycles.

Furthermore, the comparison between the estimated $q_{o,cal}$ values (calculated using the Thomas model) and the experimental $q_{o,exp}$ values shows that there was no significant

difference between them ($p > 0.05$; Anova test). This indicates that the Thomas model provided a satisfactory description of the simulated experimental data, as the calculated values closely matched the experimental values. Overall, the results suggest that the Thomas model was able to capture the decreasing adsorption capacity of the composite over desorption and regeneration cycles, and the calculated values from the model were in good agreement with the experimental values.

## 4. Comparison of This Work with Other Studies

The effectiveness of the composite in removing COD and NH$_3$-N from the leachate in a fixed-bed column was investigated, and its adsorption capabilities were compared to adsorbents from previous work. Table 7 summarizes the effectiveness of several adsorbents in corresponding column operations. Comparative studies were conducted to assess their viability in terms of bed depth (cm), flow rate (mL/min), influent concentration (mg/L), saturated time (min), breakthrough point (min), and adsorption capacity (mg/g).

**Table 7.** Summary of other adsorbents for column adsorption of various contaminants.

| Type of Adsorbent | Target Pollutant | pH | Influent Concentration (mg/L) | Flow Rate (mL/min) | Bed Depth (cm) | Breakthrough Point (min) | Satu-rated Point (min) | Adsorption Capacity (mg/g) | References |
|---|---|---|---|---|---|---|---|---|---|
| Chitosan-coated carbon composite | COD NH$_3$-N | 7.8 | 7000 1500 | 3 3 | 5 5 | 40 | 120 | 1.51 1.53 | This study |
| Granular activated carbon | Acetaminophen | 9 | 40 | 2 | 45 | 5300 | 10,800 | 38.20 | [55] |
| Alkyl benzene sulfonic acid | Detergent and COD | NA | 30 | 10 | 7.5 | 60 | 180 | 216 | [58] |
| Cow-bone-based biocomposite | COD | 7 | NA | 1.4 | 30 | 309 | 600 | NA | [59] |
| Carbon-mineral composite | NH$_3$-N COD | NA | 1640 2257 | 8 | 36 | 150 | 500 500 | 4.46 3.23 | [60] |
| *Hibiscus cannabinus kenaf.* | Cr(VI) | 7 | 0.5 | 2 | 15 | 22.5 | 90 | 21.48 | [61] |
| Goethite-modified natural sand | 3,4-dihydroxy-benzoic acid | 5 | 60 | 1 | 9.8 | 93 | 263 | 35.66 | [62] |
| MCM-41 | Methyl green | 6 | 20 | 0.8 | 6 | 250 | 1400 | 20.97 | [63] |
| Chitosan/alumina | Nitrate | 6.8 | 100 | 8 | 40 | 510 | 600 | 25.52 | [57] |

Note: NA: not available.

It is encouraging to note that the chitosan-coated carbon composite is one of the most promising adsorbents for COD and NH$_3$-N adsorption. In Table 7, different adsorbents exhibited their adsorption capacities depending on the optimum conditions of fixed-bed studies, including pH, bed depth, and flow rate. The adsorbents include granular activated carbon (38.2 mg/g), alkyl benzene sulfonic acid (216 mg/g), *Hibiscus cannabinus kenaf* (21.48 mg/g), goethite-modified natural sand (35.6 mg/g), and MCM-41 (20.9 mg/g).

Different composites have better adsorption capabilities. Halim et al. [60] reported that the carbon composite exhibited an adsorption capacity of 4.6 and 3.23 mg/g with NH$_3$-N and COD concentrations of 1640 and 2257 mg/L, respectively. Similarly, Golie et al. [57] found that a chitosan alumina composite treated 100 mg/L of nitrate with an adsorption capacity of 25.5 mg/g. In this work, the composite attained adsorption capacities of 1.51 and 1.53 mg/g for COD and NH$_3$-N, respectively. Although the adsorption of COD and NH$_3$-N by the composite can be enhanced by coating CS with chitosan, treated effluents could not meet the required COD and/NH$_3$-N limits of less than 200 and 5 mg/L, respectively, set by local legislation. This suggests that further optimization or combination with other treatment processes may be necessary to achieve the desired effluent quality.

## 5. Conclusions

The findings of this fixed-bed study indicate that the chitosan-coated coconut-shell composite has potential engineering applications for the treatment of landfill leachate. The composite showed improved performance in removing COD and NH$_3$-N compared to CS

and chitosan. The breakthrough curves obtained from the fixed-bed studies followed the expected "S" shape, indicating an efficient adsorption behavior.

The results also indicated that the breakthrough points varied among the different adsorbents, with CS having the fastest breakthrough point, followed by chitosan, and then the composite. This suggests that the composite has a higher adsorption capacity and a longer breakthrough time. The study highlighted the importance of bed depth and flow rate in optimizing column studies. Lower flow rates and deeper bed depths were found to be beneficial for achieving better removal efficiencies.

The regeneration of the composite showed promising results, with approximately 96% recovery after the first cycle, indicating its potential for multiple adsorption cycles. Although the treated effluent did not meet the COD and $NH_3$-N limits set by legislation, the use of the composite derived from unused resources holds promise for the removal of COD and $NH_3$-N from landfill leachate. Further research using a subsequent biological process, such as activated sludge, may be needed to meet the required effluent standards.

**Supplementary Materials:** The following supporting information can be downloaded at: https: //www.mdpi.com/article/10.3390/w15122263/s1, Figure S1: Characterization data of composite before column studies; Figure S2: Characterization data of composite after column studies.

**Author Contributions:** Conceptualization, A.A.; methodology, H.H.G.; validation, C.M.; formal analysis, K.W.C.; investigation, F.B.; resources, M.H.D.O.; writing—original draft preparation, T.A.K.; writing—review and editing, T.A.K.; supervision, A.M. All authors have read and agreed to the published version of the manuscript.

**Funding:** This work received grants No. Q.J130000.21A6.00P14 and No. Q.J130000.3809.22H07 from the Universiti Teknologi Malaysia (UTM).

**Data Availability Statement:** Data are available upon request.

**Conflicts of Interest:** The authors declare no conflict of interest. The funders had no role in the design of the study; in the collection, analyses, or interpretation of data; in the writing of the manuscript, or in the decision to publish the results.

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
