# Peer review of "Fixed-Bed Studies of Landfill Leachate Treatment Using Chitosan-Coated Carbon Composite"

_water, doi:10.3390/w15122263_

Round 1

Reviewer 1 Report

This manuscript reports on the use of chitosan based carbon composite to treat landfill leachate in a fixed bed system. The manuscript presents an interesting set of data, but the overall quality of the writing needs improvement before the study can be published in this journal.

1.     The logic of the abstract needs improvement.  The coconut shell in line 27 comes out of the blue.

2.     The introduction is far too long; length should be reduced by 50%.

3.     The authors state that (in its current/tested from) the fixed bed system does not meet  the effluent standards.  The authors should use the data set generated to design/engineer a system that would meet the standards. How much larger should the system be to meet these requirements? Or what flow would the bed be able to handle satisfactory?

4.     What would the critical parameter be in such a design? COD or NH3?

Author Response

Manuscript ID: water-2434735

Title: Fixed bed studies of landfill leachate treatment using chitosan-coated carbon composite

The authors are grateful to the reviewer for his/her useful feedbacks to our work. The constructive suggestions have significantly improved the quality of the manuscript after revision.

Reviewer 1

  1. Comment

The logic of the abstract needs improvement.  The coconut shell in line 27 comes out of the blue.

Reply to Comment:

As suggested, the logic of the abstract has been revised accordingly. The word “coconut shell” has mentioned from the beginning of the abstract on page 1 line 21.

“The feasibility of a chitosan-coated coconut shell (CS) composite”…..

  1. Comment

The introduction is far too long; length should be reduced by 50%.

Reply to Comment:

The introduction has been shortened accordingly. From readers’ perspective, it is beneficial to understand the logical flow of the root of the problem from downstream to its solution (upstream).

  1. Comment

The authors state that (in its current/tested from) the fixed bed system does not meet the effluent standards. The authors should use the data set generated to engineer a system that would meet the effluent standards. How much larger should the system be to meet the requirements? Or what flow would the bed be able to handle satisfactory?

Reply to Comment:

Identical experiments have been replicated for three times and gave identical results. To meet the effluent discharge standards, further treatment using biological process such as activated sludge is necessary to complement biodegradation of the wastewater samples. Please refer to the changes on page 13 lines 557-558.

Lower flow rates and deeper bed depths were found to be beneficial for achieving better removal efficiencies. Please refer to the changes on page 13 lines 551-552.

  1. Comment

What would the critical parameter be in such a design? COD or NH3?

Reply to Comment:

Critical parameters for this work included breakthrough time, flow rate, and loss of adsorption capacity throughout the cycles. Please refer to the page 6 line 268-269.

Reviewer 2 Report

This work entitled “Fixed bed studies of landfill leachate treatment using chitosan-2 coated carbon composite” reports a low cost composite for wastewater treatment. To the reviewer, the structure and results were well presented. However, there are some questions that should be addressed before a decision for publication.

1.        Table 1. Properties of CS, should be cited?

2.        There are some good manuscripts (Nature Catalysis volume 4, pages10321042 (2021)ï¼›Advanced Science, 2300841ï¼›Nature Communications volume 10, Article number: 5181 (2019); ACS Catal. 2022, 12, 22, 1415214161, and others) that may enrich the research background, suggest the authors to cite and discuss somehow in the introduction section.

3.        Additional characterization of the catalysts before and after use, such as BET, XPS, SEM-EDS etc, should be carried out.

4.        A comparison of the methane dry reforming performance between previously reported system and the present catalysts would be better to show the potential application of this candidate.

It looks fine, but some grammer mistakes and typos could by improved.

Author Response

 Manuscript ID: water-2434735

Title: Fixed bed studies of landfill leachate treatment using chitosan-coated carbon composite

First of all, the authors are grateful to the reviewer for his/her useful feedbacks to our work. The constructive suggestions have significantly improved the quality of the manuscript after revision.

Reviewer 1

  1. Comment

Table 1. Properties of CS should be cited?

Reply to Comment:

       Mistyping of the citation has been corrected on page 3 line 138.

       The properties of coconut shell (CS) and chitosan used in this study are presented in Table 1.

  1. Comment

There are good manuscripts (Nature Catalysis volume 4, pages1032–1042 (2021); Advanced Science, 2300841; Nature Communications volume 10, Article number: 5181 (2019); ACS Catal. 2022, 12, 22, 14152–14161) that may enrich the research background, suggest the authors to cite and discuss somehow in the introduction section.

Reply to Comment:

The suggested references have been incorporated in the revised manuscript on pages 16-17.

  1. Song, S.; Song, H.; Li, L.; Wang, S.; Chu, W.; Peng, K.; Meng, X.; Wang, Q.; Deng, B.; Liu, Q.; Wang, Z. A selective Au-ZnO/TiO2 hybrid photocatalyst for oxidative coupling of methane to ethane with dioxygen. Catalysis 2021, 4, 1032-42. Doi: 10.1038/s41929-021-00708-9,
  2. Li. J.; Li, L.; Ma, X.; Han, X.; Xing, C.; Qi, X.; He, R.; Arbiol, J.; Pan, H.; Zhao, J.; Deng, J. Selective ethylene glycol oxidation to formate on nickel selenide with simultaneous evolution of hydrogen. Adv. Sci. 2023, 22, 2300841. Doi: 10.1002/advs.202300841
  3. Akri, M.; Zhao, S.; Li, X.; Zang, K.; Lee, AF.; Isaacs, MA.; Xi, W.; Gangarajula, Y.; Luo, J.; Ren, Y.; Cui, YT. Atomically dispersed nickel as coke-resistant active sites for methane dry reforming. Nat. Comm. 2019, 10, 5181. Doi: 10.1038/s41467-019-12843-w
  4. Zhang, X.; Deng, J.; Lan, T.; Shen, Y.; Zhong, Q.; Ren, W.; Zhang, D. Promoting methane dry reforming over Ni catalysts via modulating surface electronic structures of BN supports by doping carbon. ACS Catal. 2022, 12, 14152-61. Doi: 10.1021/acscatal.2c04800

  1. Comment

Additional characterization of the catalysts before and after use, such as BET, XPS, SEM-EDS should be carried out.

Reply to Comment:

This work did not use any ‘catalyst’ for COD and NH3-N removal. Therefore, it is unnecessary to characterize the absent ‘catalyst’.

  1. Comment

A comparison of the methane dry reforming performance between previously reported system and the present catalysts would be better to show the potential application of this candidate.

Reply to Comment:

This work did not relate to the use of additional a catalyst in a fixed bed system for CH4 removal

Round 2

Reviewer 1 Report

The manuscript reads much better now! I have nothing further. Congratulations.

Reviewer 2 Report

By considering the positive comments from another reviewer, it can be published in the present form.